# Unusual Cell Structures and Organelles in *Giardia intestinalis* and *Trichomonas vaginalis* Are Potential Drug Targets

**DOI:** 10.3390/microorganisms10112176

**Published:** 2022-11-02

**Authors:** Marlene Benchimol, Ana Paula Gadelha, Wanderley de Souza

**Affiliations:** 1Laboratorio de Ultraestrutura Celular Hertha Meyer, Centro de Ciêcias da Saúde, Instituto de Biofísica Carlos Chagas Filho, Universidade Federal do Rio de Janeiro, Cidade Universitaria, Rio de Janeiro 96200-000, Brazil; 2Instituto de Biofísica Carlos Chagas Filho, Universidade Federal do Rio de Janeiro, Rio de Janeiro 21941-901, Brazil; 3Instituto Nacional de Ciência e Tecnologia em Biologia Estrutural e Bioimagens e Centro Nacional de Biologia Estrutural e Bioimagens, Universidade Federal do Rio de Janeiro, Rio de Janeiro 21941-901, Brazil; 4Diretoria de Metrologia Aplicada as Ciências da Vida, Instituto Nacional de Metrologia, Qualidade e Tecnologia (INMETRO), Rio de Janeiro 25250-020, Brazil; 5CMABio, Escola Superior de Saúde, Universidade do Estado do Amazonas-UEA, Manaus 69850-000, Brazil

**Keywords:** anaerobic parasites, chemotherapy, hydrogenosome, ventral disc

## Abstract

This review presents the main cell organelles and structures of two important protist parasites, *Giardia intestinalis*, and *Trichomonas vaginalis;* many are unusual and are not found in other eukaryotic cells, thus could be good candidates for new drug targets aimed at improvement of the chemotherapy of diseases caused by these eukaryotic protists. For example, in *Giardia*, the ventral disc is a specific structure to this parasite and is fundamental for the adhesion and pathogenicity to the host. In *Trichomonas*, the hydrogenosome, a double membrane-bounded organelle that produces ATP, also can be a good target. Other structures include mitosomes, ribosomes, and proteasomes. Metronidazole is the most frequent compound used to kill many anaerobic organisms, including *Giardia* and *Trichomonas*. It enters the cell by passive diffusion and needs to find a highly reductive environment to be reduced to the nitro radicals to be active. However, it provokes several side effects, and some strains present metronidazole resistance. Therefore, to improve the quality of the chemotherapy against parasitic protozoa is important to invest in the development of highly specific compounds that interfere with key steps of essential metabolic pathways or in the functional macromolecular complexes which are most often associated with cell structures and organelles.

## 1. Introduction

*T. vaginalis* and *G. intestinalis* are protist parasites causative of urogenital and intestinal infections, respectively. Trichomoniasis, caused by *T. vaginalis*, is the most common non-viral sexually transmitted infection (STI) in the world, whereas *G. intestinalis* (also named *Giardia lamblia*, *Giardia duodenalis)* is the etiologic agent of Giardiasis. This chronic condition causes one of the most common waterborne diseases, and mainly affects children. *G. intestinalis* is an extracellular flagellated protozoan parasite, class Fornicata, belonging to the order Diplomonadida, which is part of the Hexamitidae family [1]. *Giardia* has been described in most vertebrates’ intestinal tracts; however, this parasite was recently identified in the *termite gut* [2]. *Trichomonas* has a simple life cycle and belongs to the Phylum Parabasalia, order Trichomonadida.

## 2. Chemotherapy against Parasitic Protozoa

It is well-recognized that chemotherapy against diseases caused by parasitic protozoa is still based on the use of compounds developed many years ago. Although they have played a significant role during all these years of use, it is important to remember that they show variable toxicities. In addition, drug resistance to these compounds is increasing. Therefore, it is highly relevant to identify new active compounds against the pathogenic protozoa. Furthermore, repurposing drugs already used to treat unrelated diseases is an interesting alternative that needs to be stimulated.

There are several approaches to developing new chemotherapeutic agents against a parasitic protozoon. The first one, which corresponds to the most traditional way of identifying new compounds to be tested, is based on extracts and even molecules obtained from natural products of diverse origins. This approach allows for the identification of important molecules with anti-parasitic activities that are subsequently used as lead molecules to be improved by chemical synthesis, allowing drug banks to establish thousands of molecules. In addition, the test of thousands of compounds in a short time was significantly facilitated by the use of high throughput techniques. A second approach corresponds to the repurposing of already-known compounds used to treat several diseases, and whose use in humans has been approved by national authorities, as is the case with the Food and Drug Administration of the United States of America. Finally, a third approach is the design of drugs that may interfere with previously established targets. In this case, it is now possible to make theoretical previsions of their actions and subsequently proceed with the chemical synthesis of those molecules considered of high potential use. Some structures and organelles, such as hydrogenosomes in trichomonads and the ventral disc in *Giardia*, are fundamental for the parasite’s survival. Thus, drugs that can interfere with the functions played by these structures and organelles may constitute new alternatives for treating the diseases caused by this pathogenic protist. Furthermore, concerning compounds targeting specific metabolic pathways, it is necessary to consider that the microaerophilic/anaerobic parasitic protozoa present many metabolic enzymes involved in important processes, such as the triosephosphate isomerase in *Trichomonas vaginalis* encoded by two functional genes, TvTIM1 and TvTIM2. One of these is localized on the parasite surface, involves interaction with extracellular matrix and basement membrane proteins, such as laminin and fibronectin, and is involved in pathogenesis [3]. This fact opens the possibility of obtaining drugs that inhibit only this property. For example, recently, Ref. [4] identified one compound that interfered only with the non-glycolytic function of TvTIM and showed anti-*T. vaginalis* activity.

ATP generation in protozoa such as *T. vaginalis* and *Giardia intestinalis*, parasites that lack mitochondria, occurs exclusively through substrate-level phosphorylation. Sulfur-containing-amino-acid metabolism is a divergent metabolic pathway that occurs in both organisms and may constitute drug targets [5]. Fe-S-clusters play an important metabolic role in some protozoa, and there are three independent systems for their biosynthesis. In the case of *T. vaginalis* and *G. intestinalis*, only the so-called ISC system, localized in the hydrogenosomes and mitosomes, respectively, exist. This system may also constitute an important drug target, given Fe-S clusters’ key role in these organisms.

Here we will mention just a few examples of new compounds. For *G. intestinalis*, interesting results have been obtained with (a) the antibiotic fumagillin isolated from *Aspergillus fumigatus* and showed to be active even in metronidazole-resistant isolates, (b) proton pump inhibitors such as omeprazole which also inhibit triosephosphate isomerase, an important enzyme involved in the glucose and glycogen metabolism, (c) auranofin, a gold(I) complex already approved by the FDA to be used to treat rheumatoid arthritis and gold(I) phosphine derivatives, that inhibit thioredoxin reductase [6]. Another potential drug target is related to mechanisms involved in controlling gene expression, such as transcription factors [7,8] and enzymes involved in modifying histones. Examples include recent studies showing that some sirtuin inhibitors interfere with *G. intestinalis* growth, induction of multinucleated cells, and even cell death [9]. Nicotinamide also arrests *Giardia* trophozoites in the G2 phase [10].

The main focus of this review is a short description of the various structures and organelles found in the two protists, emphasizing the unique structures. They may constitute potential new drug targets. In addition, we will briefly discuss the presently used drugs for treating diseases caused by *T. vaginalis* and *G. intestinalis*.

## 3. *Trichomonas vaginalis* Features

*T. vaginalis* is an extracellular, microaerophilic protozoan that colonizes humans’ genital or urinary tracts (Figure 1, Figure 2 and Figure 3). Trichomoniasis is a sexually transmitted infection affecting approximately 400 million people worldwide [11]. Women are more symptomatic, with vaginal odor, discharge, and frequent miscarriages, which can lead to infertility [12]. Men are usually asymptomatic, although it can lead to urethritis and prostate cancer [13]. A greater predisposition to HIV (human immunodeficiency vírus) [14] and HPV (human papillomavirus) is also related to trichomoniasis [15].

*T. vaginalis* is 9 to 23 in length and 7 µm in width and can display different morphologies (Figure 2 and Figure 3). This variation will depend on its virulence level, the strain, and if there is proximity to other cells, such as epithelial cells or bacteria. Another important change in morphology occurs when the parasites are under intense stress due to nutrient deprivation or drug treatment (Figure 2c). When *T. vaginalis* are grown in an axenic medium (TYM) [16], the cells are free-swimming and pyriform. However, in vivo or when interacting with host cells, *T. vaginalis* changes its shape, displaying a variable amoeboid form with various projections to increase contact with the host cell (Figure 2b).

**Figure 1 microorganisms-10-02176-f001:**
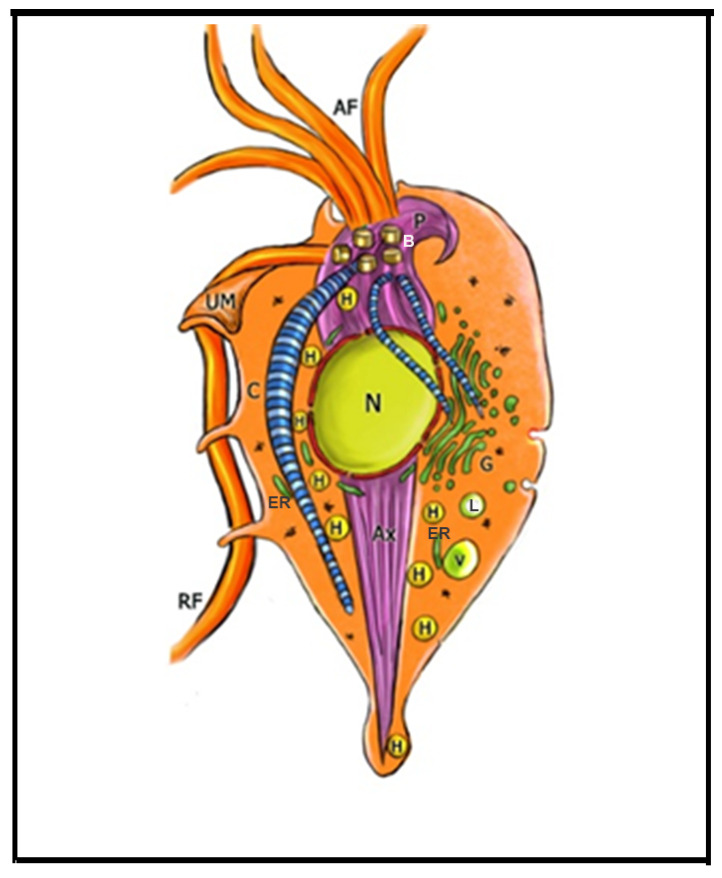
Schematic diagram of *Trichomonas vaginalis* showing the main cell structures. AF: anterior flagella; Ax: axostyle; B: basal body; C: costa; ER: endoplasmic reticulum; G: Golgi; H: hydrogenosomes; L: Lysosome; N: nucleus; P: pelta; PF: parabasal filament; RF: recurrent flagellum; UM:undulating membrane; V: vacuole. (After Ref. [17]).

**Figure 2 microorganisms-10-02176-f002:**
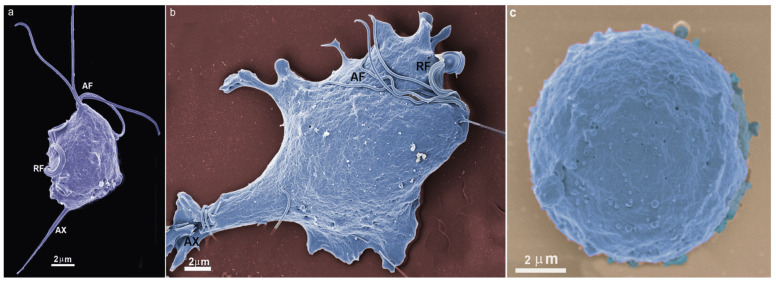
*T. vaginalis* in three different situations: (**a**) the cell was grown in an axenic medium (note the pear shape); (**b**) the parasite is adhered to a substrate and becomes flat and of a greater size; and (**c**) pseudocyst or endoflagellar; the parasite is under stressful situations; it internalizes the flagella and becomes rounded. AF, anterior flagella; RF, recurrent flagellum; Ax, axostyle. (Benchimol, unpublished).

**Figure 3 microorganisms-10-02176-f003:**
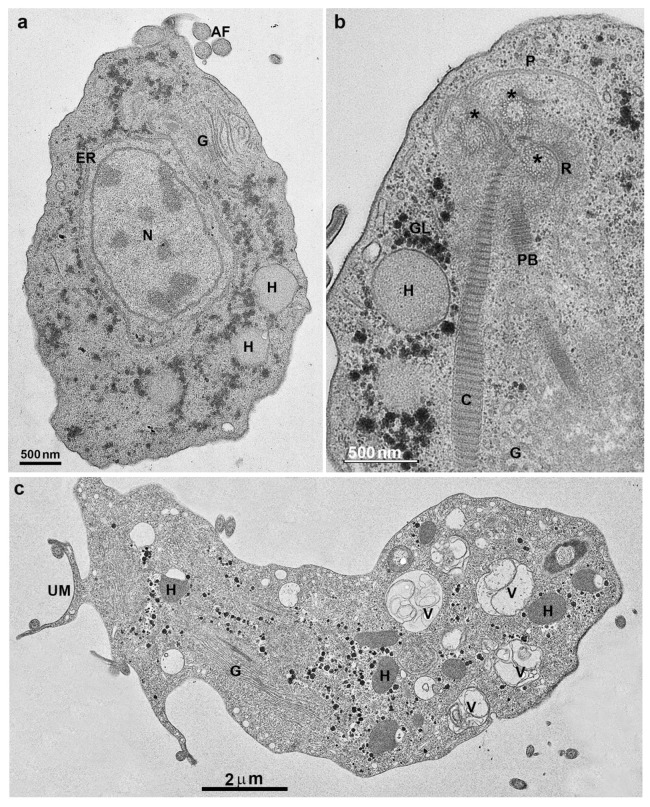
Transmission electron microscopy of *T. vaginalis*: (**a**,**b**) cells were grown in axenic media. AF, anterior flagella; N, nucleus, H, hydrogenosomes; G, Golgi; ER, endoplasmic reticulum; C, costa; P, pelta; PF, parabasal filaments; R, rootlets structures; asterisks, basal bodies. (**c**) Parasite in ameboid form. V, vacuole; UM, undulating membrane. (Benchimol, unpublished).

Trichomonads do not present the cyst form, only the trophozoite, where the flagella are externalized (Figure 1 and Figure 2a,b). However, endoflagellar forms (Figure 2c) are observed in several situations, either in samples recently obtained from infected people or by stressful induction in laboratory experiments. In this case, starvation, removal of iron from the culture medium, and addition of anti-mitotic drugs induce the internalization of flagella, which are kept in cell vacuoles, and the parasite takes a rounded shape (Figure 2c). The flagella continue beating inside intracellular vacuoles [18]. This form, named pseudocyst, or endoflagellar form, is observed in large numbers and is generally reversible when the stressful situation is eliminated.

Trichomonads present one anterior nucleus (Figure 1 and Figure 3a) with a random distribution and six chromosomes which are conservated in size and number among isolates [19]. Its mitosis occurs without nuclear envelope breakdown (closed mitosis) and with an extranuclear spindle (paradesmosis) [20].

*T. vaginalis* has five flagella, four anterior, and one recurrent flagellum forming the undulating membrane. The basal bodies are in the most parasite anterior region, where the flagella emerge through the flagellar canal (Figure 1, Figure 2 and Figure 3). The recurrent flagellum projects toward the posterior cell region, contacting the cell’s plasma membrane and forming the undulating membrane. It has been observed by deep etching that there are filamentous bridges connecting the cell surface with the recurrent flagellum [21]. All flagella participate in the cell’s movement.

The axonemes of these flagella are typically eukaryotic with a 9 + 2 array of microtubules. By freeze-fracture techniques, the anterior flagella display an impressive array of rosettes formed by 9–12 intramembranous particles [22]. In contrast, the base of the recurrent flagellum exhibits a flagellar necklace with a distinct array of particles [23].

The basal bodies in trichomonads are atypical structures since they are associated with several filaments and striated roots in hook-shaped lamellae (Figure 3b). These fibers can be contractile and noncontractile, such as the costa [24].

Trichomonads contain organelles that are common to all eukaryotic cells, such as the Golgi complex, endoplasmic reticulum, one nucleus, lysosomes, and a complex cytoskeleton forming the mastigont system (Figure 3). Several unique structures form the mastigont system, such as the pelta and the axostyle, which form the pelta axostylar complex, the parabasal filaments, and the costa (Figure 3b).

The rough endoplasmic reticulum in trichomonads is around the nucleus, in the outer nuclear membrane, and dispersed in the cell’s cytosol (Figure 1 and Figure 3a). The ER participates in the autophagic process, enlarging when the parasite is submitted to drug treatment [25]. *T. vaginalis* present a large Golgi complex (Figure 1 and Figure 3) and has been named Parabasal Apparatus, which includes the parabasal filaments [26].

*T. vaginalis* shows intense phagocytic activity, incorporating bacteria and various particles, forming large vacuoles (Figure 3c). The endocytosed material travels to lysosomes, where it is digested.

The pelta is made of stable microtubules and involves the flagellar canal from which the flagella emerge; it overlaps with the axostyle microtubules forming the pelta-axostylar system [27,28] (Figure 4). The axostyle is formed by a well-organized array of stable microtubules extending across the length of the cell and is a supportive entity (Figure 5b). In addition, it participates in trichomonads mitosis, providing constriction of the nucleus during karyokinesis.

## 4. *Trichomonas* and Its Unusual Structures

Trichomonads’ cytoskeleton presents other unusual proteinaceous structures such as some filaments and lamellae as the sigmoidal filaments, the supra- and infra-kinetosomal bodies, rootlets fibers, and striated fibers as the costa and parabasal filaments [26].

The costa (Figure 3b and Figure 4) is a proteinaceous, periodic structure placed along the cell, dissipating the stress caused by the beating of the recurrent flagellum [17,28,29]. Fine fibers connect the undulating membrane to the cytoplasmic side, where the costa is found (Benchimol et al., 1993). The costa size reaches a length of about 14.38 µm and 36.5 nm wide with alternating bands of 13.8 nm and 241.3 nm in width [30].

The costa is a non-motile structure and presents many proteins, some of which are uncharacterized. In addition, a costa accessory structure has been demonstrated (Figure 4) [30]. Further analyses of the costa fraction identified 54 hypothetical proteins, with fourteen proteins as the fraction’s major components. Thus, the costa structure presents a new class of proteins not described in other cells. More recently, one major protein (*T. foetus* ARM 19800.1 protein) was characterized and localized in the costa and designated as costain 1 [31].

### Hydrogenosomes

*T. vaginalis* does not have mitochondria but contains the hydrogenosome (Figure 5), an unusual organelle surrounded by two closely apposed membranes; it has this name because it produces molecular hydrogen.

Hydrogenosomes are considered divergent forms of mitochondria adapted to anaerobic life. In trichomonads, the hydrogenosome has 0.5 µm in diameter, is usually spherical, and may contain a peripheral, flattened, membrane-bounded compartment (Figure 5a,b) that contains high calcium levels, magnesium, and phosphorus, possibly functioning in the regulation of intracellular calcium [32,33,34]. Thus, all evidence points to the organelle as the primary calcium storage in *Trichomonas*.

The hydrogenosome participates in the pyruvate metabolism and produces ATP and molecular hydrogen (Figure 6) [35]. Since two membranes coat the hydrogenosome, it divides like mitochondria [36], imports proteins post-translationally [37], and produces ATP [38], it has been considered a modified mitochondrion. However, it does not have a genome, Krebs cycle, or the typical membranous respiratory chain. In addition, the hydrogenosome also lacks the F0–F1 ATPase, cytochromes, and oxidative phosphorylation [39]. Hydrogenosomes use the pyruvate or malate to acetate for ATP production and produce molecular hydrogen using substrate-level phosphorylation [40]. One important characteristic of hydrogenosomes is the presence of cardiolipin (de [41], a bacterial and mitochondria membrane phospholipid suggesting its endosymbiotic origin.

Previous work [42] demonstrated that hydrogenosomes and mitochondria present common core membrane components, which are important for protein import and metabolite exchange. In addition, many proteins have been localized in the matrix of hydrogenosomes, and enzymes responsible for iron–sulfur (Fe–S) cluster assembly have been localized in the *T. vaginalis* hydrogenosome [43,44].

Previous studies provided a detailed proteomic analysis of the *T. vaginalis* hydrogenosome and showed that it contains 569 proteins [45]. In addition, the authors found many proteins that function in energy and amino acid metabolism, flavin-mediated catalysis, Fe-S cluster formation, membrane translocation, oxygen stress response, proteolytic processing, chaperonin activities, and ATP hydrolysis, which are responsible for ~30% of the hydrogenosome proteome.

Recently, an overview of various aspects of this organelle, such as its biogenesis, hydrogenosomal protein import, and membrane translocases [46].

Hydrogenosomes participate in various protein synthesis, including components localized in the outer and inner membrane, and transported into the organelle using elaborated import machinery that presents some similarities to the system found in mitochondria. Several enzymes include processing peptidases, adenylate kinase, acetate: succinate CoA transferase, hydrogenase, pyruvate: ferredoxin oxidoreductase, superoxide dismutase, and several others that are involved in metabolic activity.

## 5. Drugs Affecting Trichomonas

### 5.1. Metronidazole

Metronidazole and other nitroimidazoles have been used in trichomoniasis treatment. Two important problems in trichomoniasis treatment using metronidazole are the resistance and side effects, such as metallic taste, vomiting, nausea, dizziness, and insomnia [47]. In some cases, it provokes leucopenia and neuropathies. Moreover, this drug is prohibited during pregnancy [48], and some strains exhibit 5’-nitroimidazoles resistance. In addition, several hydrogenosomal proteins are altered in drug resistance, resulting in severe organelle modifications [49]. The authors noted a marked reduction of pyruvate: ferredoxin oxidoreductase and ferredoxin levels in resistant strains. Furthermore, one group [50] described a different pathway involved in the metronidazole activation within the hydrogenosome. They reported that trichomonads acquired a high level of metronidazole resistance when both the pathways of malate and pyruvate that activate metronidazole were eliminated.

Metronidazole enters the cell as an inactive prodrug by simple diffusion and goes to the hydrogenosome using the same way [51]. It needs to find a highly reductive environment to be reduced to the nitro radicals to be active. Oxidoreductases like pyruvate ferredoxin oxide reductases make such a reduction. Major oxygen-scavenging enzymes include Flavin reductase and NADH oxidase. For instance -N(2-hydroxyethyl) oxamic acid and acetamide may damage the DNA of replicating cells. The mode-of-action of the heterocyclic aromatic nitro-compounds is considered to be due to the radical damage caused by the reactive and toxic species that are obtained from the reduction of the nitro groups and that interact with several intracellular molecules, including DNA [52,53]. Resistance of the parasites to metronidazole reaches around 4.3% of the isolates in the USA [54]. Hydrogenosomes are the main target for activating 5-nitroimidazole drugs [55]. In the hydrogenosome (Figure 7), metronidazole is activated to a cytotoxic form. In addition, morphological studies provided evidence for hydrogenosome alterations in size, shape, and behavior when these drugs are used in vitro [49,56].

### 5.2. Effects on Trichomonas Structures by Other Drugs (Table 1)

The plasma membrane, the endoplasmic reticulum (ER), and the Golgi complex are key components of the cells. They may present unique components involved in the capacity of the protozoa to interact with host cells. For *Trichomonas* and *Giardia*, there is evidence of some proteins exposed on the protozoan surface or secreted via extracellular vesicles (exosomes, ectosomes) that are fundamental for the protozoan to exert its pathogenic side. Examples include cysteine proteases and variable surface proteins (VSP), some of which, as in the case of *G. intestinalis*, are involved in antigenic variation. The synthesis and the fate of these molecules to the cell surface involve the participation of intracellular vesicles and the cytoskeleton components used to transport them mediated by dyneins and kinesins. Interruption of these processes may interfere with parasite viability; therefore, they constitute molecular targets that can be interfered with by specific molecules. Indeed, there are several examples where parasites become inviable when the synthesis of a certain protein is blocked. Thus, this is an exciting area for further development of anti-parasitic drugs.

The lipidic component of the cell membranes is also a drug target explored for some parasitic protozoa. For example, it has been shown that the introduction of carbocyclic rings in the lipid portion of alkyl phosphocholines leads to drugs with potent activity against some protozoa. Miltefosine is an alkylphosphocholine synthetic lipid analog shown to present activity in cancer cells and parasites protozoa such as *Leishmania, T. cruzi*, and *T. vaginalis* [57]. It is nowadays used in clinics by oral via. Other ether phospholipid derivatives were synthesized and showed improved activity and lower toxicity against parasites tested for *T. vaginalis* [58,59,60]. Using miltefosine (MLT), several alterations, such as wrinkled and rounded cells, membrane blebbing, intense vacuolization, and nuclear condensation, occurred, all indicative of cell death by apoptosis (Figure 8) [57]. In addition, cells treated with the IC50 of MLT significantly reduced the number of viable parasites. One group [61] used clotrimazole (CTZ) and zinc compounds, and CTZ complexed with zinc salts acetate [Zn(CTZ)2(Ac) and a chloride [Zn(CTZ)2Cl2] complexes against *T. vaginalis*. The incubation of the parasites with [Zn(CTZ)2(Ac)2] complex inhibited parasite growth and provoked changes in the shape of treated parasites with cell membrane projections. In addition, hydrogenosomes, endoplasmic reticulum, and Golgi complex were altered (Figure 9). Therefore, the authors inferred that [Zn(CTZ)2(Ac)2] is a highly effective compound against *T. vaginalis* in vitro, indicating its potential alternative use as an agent against trichomoniasis. *T. vaginalis* treated with [Zn(CTZ)2(Ac)2] exhibited a reduction of pyruvate: ferredoxin oxidoreductase in hydrogenosomes.

It has been reported cell alterations provoked by Δ(24(25))-sterol methyltransferase inhibitors on *T. vaginalis* [47]. The authors described parasites forming cell clusters with wrinkled cells, membrane blebbing, and cell disruption. In addition, the Golgi was abnormal, the hydrogenosomes were damaged, and several autophagic vacuoles were observed [47]. When a comparative study was performed using metronidazole and BPQ-OH [62], both drugs provoked cell death, intense cell vacuolization, and membrane blebbing. However, BPQ-OH was less toxic for human cells in vitro.

One group reported that the compounds amioder and dronedarone affected *T. vaginalis* [63]. The parasites were killed by an apoptosis-like process and showed morphological changes with disturbance in the hydrogenosome structure (Table 1).

### 5.3. Proteasomes

The proteasome is a macromolecular complex that controls exceeding protein formation and erroneous proteins, acting on its degradation. Thus, cell homeostasis can be maintained due to its proteolytic activities involving breaking the peptide bonds. It has been reported that the protist parasite *Tritrichomonas foetus*, a relative of *T. vaginalis*, presents a 20S proteasome [64]. The proteins are tagged via a single ubiquitin molecule with the enzyme ubiquitin ligase. Any interference with this system causes severe damage to the cell. Therefore, it may constitute an important drug target. For instance, drugs such as Ixazomib and carmaphycin-17, which interfere with proteasomes, are effective against *T. vaginalis* [65]. In addition, the fungal metabolite gliotoxin inhibits proteasome proteolytic activity. As a result, it induces an irreversible pseudocysts transformation (endoflagellar form (EFF) and cell death in *Tritrichomonas foetus*, a protist cattle parasite [66]). Lactacystin, a well-known specific proteasome inhibitor, interfered in transforming *T. foetus* to endoflagellar form (EFF) in a dose-dependent manner. Lactacystin treatment also resulted in an accumulation of ubiquitinated proteins. Furthermore, it caused an increase in the number of endoplasmic reticulum membranes in the parasite, thus suggesting that the ubiquitin-proteasome pathway is required for the cell cycle and EFF transformation in *T. foetus* [64].

**Table 1 microorganisms-10-02176-t001:** Drugs affecting *T. vaginalis* and their effects on cell organelles.

Drug	Cell Structure Affected	Effect	Reference
Miltefosine	Plasma membrane; nucleus; hydrogenosomes; flagella	Cell clusters; membrane blebbing; wrinkled cells; pseudocysts formation; myelin-like figures; condensed chromatin; vacuolization; decreased size of hydrogenosomes	[57]
Methyl jasmonate	Hydrogenosome	cell death; loss of hydrogenosomal membrane potential	[67]
Δ(24(25))-sterol methyltransferase inhibitors	Golgi; Hydrogenosomes; plasma membrane	cell clusters; wrinkled cells; membrane blebbing; cell disruption. abnormal Golgi; damaged hydrogenosomes; autophagic vacuoles	[47]
3-(biphenyl-4-yl)-3-hydoxyquinuclidine (BPQ-OH)	Plasma membrane; vacuoles	inhibits in vitro proliferation; rounded and wrinkled cells, membrane blebbing; intense vacuolization, o cell death; cell death	[62]
Lactacystin	ER	increase in the number of endoplasmic reticulum membranes	[64]
Zinc-clotrimazole complexes	ER; Golgi; plasma membrane	atypical ER; a high number of filopodia; irregular distribution of Golgi lamellas	[61]
Amiodarone	Nucleus; hydrogenosome	Cytokinesis blockage; duplicated cell structures; large multinucleated cells; Hydrogenosome alteration; glycogen disturbance	[63]
Amioder and Dronedarone	Hydrogenosome; plasma membrane	Apoptosis; deformed andaggregated cells; hydrogenosomes alterations	[63]
Thiosemicarbazones 49, 51, and 63	Structural cell changes;Hydrogenosomes are not affected	Pseudocysts, cell surface alterations	[68]

Recently Ibánez-Escribano et al. [68] designed and synthesized several thiosemicarbazones (Schiff-based analogs) with modification of the stereoelectronic effects of the substituents on N-1 and N-4. In addition, the authors made an isosteric replacement of sulfur with selenium in some of them. Thiosemicarbazones (numbers 49, 51, and 63) showed high activity against *T. vaginalis* with IC50 of 16.39, 14.84, and 14.89 µM, respectively. No significant cytotoxic activity against Vero cells was observed (CC50 of 256 µM). On the other hand, selenoisosters 74 and 75 showed IC50 in the range of 11 µM. These compounds did not interfere with the hydrogenosome membrane potential as analyzed by labeling with the fluorescent dye JC-1. Scanning electron microscopy of drug-treated *T. vaginalis* showed the presence of rounded cells, some resembling pseudocysts, surface invaginations, and pores.

## 6. *Giardia intestinalis*

*G. intestinalis* colonize the host’s small intestine, causing the disease known as Giardiasis. In many individuals, the infection remains asymptomatic, while others present weight loss, colic, and diarrhea. In addition, pathogenicity may vary according to the parasite strain and host-immune system [69].

*Giardia* has two forms in its life cycle: the cyst, the infective form, and the trophozoite. The infection begins when a person ingests contaminated water or food containing *Giardia* cysts. Other modes of transmission are from person-to-person contact through the fecal-oral route. During cyst passage through the stomach, it is exposed to acidic pH and digestive enzymes, which trigger the excystation. In the small intestine, each quadrinuclear cyst release two binuclear trophozoites, which proliferate and adhere to the intestinal epithelium. Trophozoites are noninvasive but cause structural and functional damage in the host cell, leading to inadequate nutrient absorption by the intestine. The encystation starts when trophozoites are transported through the intestinal lumen to the large intestine, a slightly alkaline environment containing pancreatic proteases. This process culminates with the formation of new cysts excreted with the feces [69]. The cyst wall protects the organism against adverse environmental conditions such as desiccation, cold or chemical treatment.

*Giardia* has been considered a primitive eukaryotic organism based on the sequence-based comparison using different genes (rRNA, HSP70, and cathepsin B) and ultrastructure observations. The parasite does not have mitochondria and Golgi complex and has enzymes similar to those of prokaryotes. In addition, it does not have enzymatic components of mitochondrial respiration. However, identifying genes of mitochondrial origin led to the suggestion that this organism probably lost mitochondria during evolution [70]. Furthermore, the discovery of *Giardia* which are double-membrane organelles containing proteins involved in the biogenesis of iron–sulfur clusters [71], supports the above hypothesis. These findings, together with other data, indicate that this protozoan might diverge after the endosymbiotic event resulting in the establishment of mitochondria [72].

### 6.1. Giardia intestinalis Features

*G. intestinalis* trophozoites have a pear-shaped body with a wider anterior region (Figure 10). The parasites have bilateral symmetry and are around 15 µm long and 5 µm wide. The presence of four pairs of flagella and the ventrolateral flange can be observed by scanning electron microscopy (Figure 11a). The two ovoid nuclei are easily identified in the anterior region of the cell (Figure 11b) [73]. On the ventral surface is observed the ventral disc, a microtubular structure associated with the parasite’s attachment to the intestinal epithelium [74]. The median body, another microtubular structure, is also observed in stained preparations for brightfield microscopy and electron microscopy observations (Figure 11b) [75]. Organelles such as the endoplasmic reticulum and peripheral vesicles, which represent the endosomal-lysosomal system, are also found in these organisms [76]. A typical Golgi complex was not identified in the parasite, although Golgi-like compartments have been described during trophozoite-cyst differentiation [77].

### 6.2. Giardia Cyst

*G. intestinalis* cysts measure 11–14 × 7^−10^ µm and can be round or oval (Figure 11c). They have four nuclei and contain axonemes and median bodies. The ventral disc is seen fragmented in the cytoplasm (Figure 11d). The cysts are coated by a birefringent wall of 0.3–0.5 µm thickness. The flagella are internalized and positioned in the peritrophic space (Figure 11d).

### 6.3. Giardia and Unusual Cell Structures

*G. intestinalis* cytoskeleton is organized in microtubular structures such as the ventral disc, flagella, median body, and the funis, important structures in adhesion, division, and motility [78]. Some of these cytoskeletal structures are unique for this parasite and essential for its survival and pathogenicity.

#### 6.3.1. Ventral Disc

The ventral disc is formed by microtubules arranged in a spiral (Figure 12a) and is adjacent to the plasma membrane. Trilaminar fibrous microribbons extend from the microtubule wall into the cytoplasm and are connected by cross-bridges [79]. Surrounding the disc is a dense fibrous structure known as a lateral crest [80]. The ventral disc is a structure of *Giardia*, not seen in other eukaryotes, and is considered responsible, at least in part, for the adhesion of the parasite to host cells [81]; it also participates in nuclei division [82].

For many years, the ventral disc was considered a homogeneous structure without identifying subdomains responsible for a defined structural organization. However, electron tomography and high-resolution scanning electron microscopy observations showed that its elements (microtubules, microribbons, and cross-bridges) could have a different arrangement depending on the disc’s region [79,83]. Thus, some domains were clearly defined as disc margin, ventral groove region, and overlap zone.

Tubulin is the main protein in the ventral disc proteomic analysis [84]. They can present post-translational modifications such as acetylation and polyglycylation [79]. Another abundant protein in the disc is β-giardin, identified as a major component of the microribbons and unique to the parasite. Proteomics detected other proteins using the isolated disc [84,85]. CRISPRi-mediated knockdown was performed to infer the functional role of some of these new disc-associated proteins [86].

#### 6.3.2. Median Body

The median body is a component of the trophozoite cytoskeleton localized dorsally and posteriorly to the ventral disc (Figure 12b). A variable number of layers forms it; its position and size vary in the cell [75].

#### 6.3.3. Flagella

*G. intestinalis* has four pairs of flagella that originate from basal bodies. The flagellar structure is similar to the flagella of other eukaryotes (Figure 13a,b). The eight flagella are arranged in pairs and differ in position in the cell, receiving specific names: anterior, postero-lateral, ventral, and caudal (Figure 11a). The flagella participate in parasite motility and adhesion to the host cell and play an important role in cell division [87,88].

Interestingly, the flagella of *G. intestinalis* present extra-axonemal structures, distinct in each flagellar pair [78]. For example, the marginal plates and dense rods are coupled to axonemes of the anterior flagella [89]. The funis are bands of interconnected microtubules surrounding and extending dorsally and ventrally to the caudal axonemes. In the posterior region of the cell, some microtubules from each band spread in the direction of the postero-lateral axonemes or plasma membrane. It has been suggested that the funis participate in the caudal movements of the parasite [90].

**Figure 13 microorganisms-10-02176-f013:**
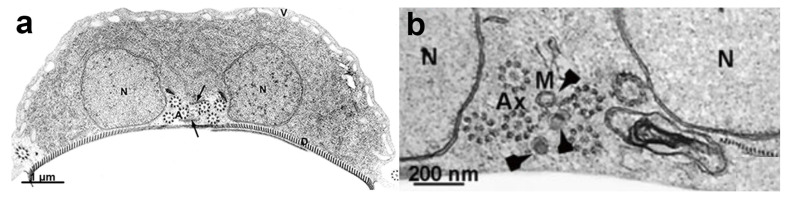
Transmission electron microscopy of *G. intestinalis* trophozoites: (**a**) two nuclei (N), peripheral vesicles (V), ventral disc (D), and funis (arrows) are observed; and (**b**) central mitosomes (arrowheads) are seen between the axonemes. (**a**) (After Ref. [90]), (**b**) (After Ref. [78]).

#### 6.3.4. Mitosomes

*G. intestinalis* does not have mitochondria but has several spherical structures bounded by a double membrane named mitosomes (Figure 13b). Although abundant in the parasite, the presence of this organelle was only demonstrated in 2003 [71]. Mitosomes are considered vestigial mitochondrion and probably perform functions lost during evolution [71]. Although described as oval compartments, [91] showed that mitosomes have asymmetric shapes and present internal compartmentalization. They measure about 150–200 nm and are found between the two nuclei and the axonemes of the flagella (central mitosomes) (Figure 14) and are also distributed throughout the cytoplasm (peripheral mitosomes) [71,92]. In addition, mitosomes lack classic mitochondrial functions, such as ATP synthesis, lipid metabolism, and the citric acid cycle. DNA was not detected. However, iron-sulfur complex biosynthesis and TIM and TOM protein import system characteristics of mitochondria are preserved in mitosomes. These organelles also have Cpn60, HSP70, and ferredoxin [71,92,93].

## 7. Endocytic and Exocytic System

*G. intestinalis* has just below its plasma membrane numerous vacuoles named peripheral vesicles (Figure 13a), which measure 100–200 nm in length and participate in the endocytic process in the parasite [76]. The endoplasmic reticulum is an endomembranous tubulovesicular network, a pluripotent organelle performing endocytosis and protein synthesis in the parasite [94]. Previous studies showed that acid phosphatase and cysteine proteases located within the peripheral vesicles are secreted to the outside, suggesting the participation of these organelles also in the process of exocytosis of the parasite [95,96]. In addition, intraluminal vesicles inside peripheral vesicles exhibited morphological characteristics of multivesicular bodies, which would be released during parasite differentiation [96].

## 8. Peroxisomes in *Giardia*

Peroxisomes are related to lipid metabolism and detoxification of reactive oxygen species. Although *Giardia* was considered an organism that lost peroxisomes during evolution, new and interesting findings were reported [97]. These authors observed 0.1–0.5 µm vesicles in the parasite, which were positive for 3,3′-diaminobenzidine (DAB), a classic morphological marker used to detect enzymatic activities in peroxisomes. Furthermore, two peroxisomal proteins as, peroxin-4 and acyl-CoA synthetase long-chain family member 4, were localized in these vesicles [97].

## 9. Nuclei

*G. intestinalis* has two transcriptionally active nuclei localized in the anterior portion of the cell (Figure 11b). They are round or oval and have the same size (Figure 14). Observations of membrane-extracted cells by electron microscopy show that the nuclei are anchored by proteinaceous links to microtubular structures in the cell, such as the basal bodies [73]. The presence of five chromosomes in each Giardia nucleus has been reported, which has a diploid genomic organization and, thus, gives a tetraploid content to the parasite [98]. The nuclear envelope with an inner and an outer membrane covered with ribosomes is quite evident in electron microscopy preparations [99]. The protozoan presents a nucleolus, and new data were obtained by [100], who identified this structure using silver and immunostaining techniques for various nucleolar markers.

*G. intestinalis* nuclei can be targets of metronidazole, a 5-nitroimidazole drug traditionally used against Giardiasis. This compound leads to the phosphorylation of histone H2A and induces DNA fragmentation [101]. Despite these effects, the authors reported that DNA was one of the targets of this compound, along with other biomolecules. Furthermore, the nuclear structure of the trophozoites was seen to alter after exposure to benzopyrrolizidine molecules. The discontinuity of the nuclear envelope, the presence of lamellar bodies in the nuclei, and the condensation of chromatin were observed in the treated cells with these compounds [102]. Intriguingly, the loss of one of the nuclei was also reported. Nuclear changes similar to those described above were also reported after incubating the trophozoites with 3-arylideneindolin-2-one-type sirtuin inhibitor, emphasizing class III histone deacetylases as an interesting target for developing anti-giardia drugs [9].

### Mitosis

*G. intestinalis* mitosis is semi-open, and two extranuclear spindles are observed [103]. The parasite nuclear envelope remains intact throughout the division process, except for the pores through which the spindle microtubules penetrate. In cytokinesis, a contractile ring is absent, and flagellar movements could contribute to cell cleavage longitudinally [104].

## 10. Drugs Affecting *Giardia*

### 10.1. Metronidazole, Albendazole and Nitazoxanide

Giardiasis treatment is based on compounds derived from the nitroimidazole class, and metronidazole is the main compound used. This compound is administrated two-three times a day for up seven days. Although it has about 73 to 100% efficacy, the increased clinical resistance and the several side effects lead to treatment failure [105]. Metronidazole causes headaches, vertigo, nausea, vomiting, and a metallic taste. In addition, its prolonged use can lead to pancreatitis, central nervous system toxicity, and peripheral neuropathy [105].

Furthermore, USA Food and Drug Administration pointed out that metronidazole presents carcinogenic activity in rats and mice. Because nitro compounds have been used in the clinic for more than 50 years, the selection of drug-resistant *G. intestinalis* can occur [106]. Metronidazole penetrates the parasite by passive diffusion, and it is activated by enzymatic reduction via pyruvate-ferredoxin oxidoreductase, nitroreductase 1, and thioredoxin reductase [106]. After reduction, the 5-nitro group generates nitro radical anions and other compounds, including nitroso and hydroxylamine derivatives, which can interact with DNA, damaging and disturbing this molecule’s duplication [101]. Consequently, changes in the parasite cycle occur with the accumulation of cells in the S phase [101]. In addition, trophozoites treated with this compound present peripheral vesicles increase and myelin-like figures [107].

The second drug used for treating Giardiasis is Albendazole, a benzimidazole carbamate compound. Treatment with this compound can be performed in a single dose for up to 7 days, provoking side effects such as nausea, vomiting, diarrhea, and abdominal pain [106]. Albendazole has an efficacy of 35–96%; high rates are reached when combined with other drugs. The USA Food and Drug Administration recommends caution in using this compound in women of childbearing age because it can be teratogenic. Albendazole binds to β-tubulin, inhibiting its polymerization in microtubules [108].

Consequently, the ventral disc is disrupted, and the parasite detaches from the substrate [109,110]. Albendazole also induces the accumulation of cells in the G2 phase, probably because it interferes with mitotic spindle formation [101]. This compound does not affect the axoneme’s or median body’s microtubules despite disrupting ventral disc microtubules [109,110]. Albendazole and mebendazole, other benzimidazole carbamate compounds, have an activity 30–50-fold higher than metronidazole or tinidazole against Giardia. Mebendazole also damages the ventral disc of *Giardia* [110].

More recently, nitazoxanide, a nitrothiazole antibiotic, was indicated to treat Giardiasis. This drug is performed three times a day and has side effects such as headaches and gastrointestinal disorders. However, it displays cure rates of 70–80% [106]. Two mechanisms of action were proposed for this compound: (1) inhibition of enzymes such as ferredoxin oxidoreductase and the nitroreductase G1NR1 of the parasite or (2) production of nitro radical anions due to the reduction of 5-nitro group contained in the molecule [111]. In vitro assays showed that after exposure to nitazoxanide or N-methyl-1H-benzimidazole hybrid CMC-20 occurred, the disruption of the parasite plasma membrane and ventral disc damage was also observed [112]. Recently, it has been shown that nitazoxanide and metronidazole led to alterations in the intestinal microbiota in rats and mice [113]. This feature could be related to the treatment’s effectiveness and prolonging the infection.

### 10.2. Effects on Giardia Structures by Other Drugs (Table 2)

Several drugs affecting *G. intestinalis* and their effects on cell components are in Table 2. It is known that compounds from different classes can lead to cell death phenotypes. Typical morphological features of an apoptosis-like process, such as chromatin condensation, cytoplasm vacuolization, and membrane blebbing, were observed after incubation with beta-lapachone, an ortho-naphthoquinone (Figure 15a,b) [114]. Similar changes were observed after incubation with curcumin [115]. Activation of *G. intestinalis* autophagy has also been described after exposure to drugs. The main characteristic observed in these cells is the presence of myelinic figures involving cytoplasmic contents or around some organelle. This phenotype was seen after incubation with 4-[(10H-phenothiazin-10-yl)methyl]-N-hydroxybenzamide, a class I histone deacetylase inhibitor (Figure 16a) [116]. Myelinic figures were also observed after exposure to metronidazole, nitrofuran furazolidone, bisphosphonates (risedronate and ibandronate), which inhibit farnesyl diphosphate synthase and 24,25-(R,S)-epiminolanosterol, a sterol biosynthesis inhibitor [107,117,118].

In addition to albendazole, nocodazole and colchicine disrupt the ventral disc microtubules, altering trophozoite adhesion to the host cell (Figure 16b) [119]. There are few studies showing drug effects on *Giardia’s* median body. Since microtubules form it, it was expected that compounds that target the microtubular dynamics would interfere with this structure. However, albendazole does not affect the median body [109]. However, nocodazole reduced microtubules in the median body, whereas Taxol increased its size [119,120].

Concerning the flagellar length, nocodazole increases flagellar length, whereas Taxol, another microtubule-target agent, decreases [120]. Cells treated with oryzalin, a nitroaniline herbicide that binds to tubulin, displayed flagella bending [121]. Kinase inhibitors also alter the flagella size. For example, after incubating trophozoites with polo-like kinase-specific inhibitors, cells showed an increase in the caudal and anterior flagella length [122]. Cytochalasin D, latrunculin A, or jasplakinolide, which interfere with actin dynamics, induce damage to the caudal region and flagella internalization [123]. After exposure to pyrantel pamoate, trophozoites decreased flagella beating, but damage in their structures was not observed [124].

*G. intestinalis* division is altered by cytochalasin, which binds actin, inhibiting the polymerization and the elongation of microfilaments. Incubation with this compound led to multiflagellated unshaped cells, indicating that the parasite does not complete cytokinesis (Figure 17a,b) [125]. In addition, 4-[(10H-phenothiazin-10-yl)methyl]-N-hydroxybenzamide, small gatekeeper kinases, and aurora kinase inhibitors induced an increase in multinucleated parasites, suggesting that these compounds can also interfere in the late-phase of the division process [116,126,127].

As commented on above, the process of division of *Giardia* and *Trichomonas* trophozoites presents specific features that make this system a potential target for developing new anti-parasitic compounds. One important step of cell division in animal cells, fungi, and the amoeba is cytokinesis, which is finalized by the constriction of an actomyosin contractile ring made of actin filaments, myosin II, and associated proteins. However, other organisms lack myosin II and use other mechanisms for cytokinesis. These organisms, which include *Giardia* and *Trichomonas*, use flagellar motility and microtubule remodeling for this process. One good example is the ventral disc’s participation in *Giardia*’s division [82]. Although typical actin filaments have not been shown in *Giardia*, it contains highly divergent actin that retains conserved roles in processes such as membrane trafficking and cytokinesis [128,129]. In addition, *Trichomonas* structures, such as the axostylar microtubules and the flagellum, play some role in cell division [20]. Several cytokinesis regulators, like CDK1-like kinases, aurora kinases, and polo-like kinases, are involved in these processes [130]. A better understanding of the molecular basis of all the processes may point to interesting drug targets that may open new chemotherapeutic possibilities. For instance, it was shown that aurora kinase inhibitors block the completion of the cytokinesis of *Giardia* [127].

Pharmacological compounds targeting the peripheral vesicles or the endocytosis/exocytosis process of *G. intestinalis* have not yet been reported. However, trophozoites increased peripheral vesicle size after exposure to metronidazole, pyrantel pamoate, and 24,25-(R,S)-epiminolanosterol (Figure 18a,b) [117,124]. However, the mechanism responsible for this effect is not known. *G. intestinalis* endocytosis is affected by cytochalasin D and anti-microtubule agents such as vinorelbine, which interfere with ceramide absorption. This process was not changed after exposure to filipin, a classic endocytosis inhibitor, suggesting that lipid rafts did not participate in this process [131].

Although the mitosome is an interesting target for drug development, structural changes in this organelle after exposure to chemical compounds have not been described. Interestingly, a cycle of nucleotide-specific phosphodiesterase has been suggested to localize in mitosomes, and inhibitors of this enzyme alter parasite proliferation [132]. In addition, glycerol-3-phosphate dehydrogenase has been shown to accumulate in the mitosome during encystation, and its activity is altered by the antitumor compound 6-(7-nitro-2,1,3-benzoxadiazol-4-ylthio) hexanol [133].

Compounds interfering with *G. intestinalis* peroxisomes have not yet been described.

**Table 2 microorganisms-10-02176-t002:** Drugs affecting *G. intestinalis* and their effects on cell organelles.

Drug	Cell Structure Affected	Effect	Reference
Metronidazole	Nuclei Vacuoles	DNA damage; accumulation of cells in the G2 phase; myelin-like figures	[101,107]
Albendazole	Ventral disc Mitosis	ventral disc disruption; accumulation of cells in the G2 phase;	[109,110]
Mebendazole	Ventral disc	ventral disc disrupted	[110]
Nitazoxanide and CMC-20	Plasma membrane Ventral disc	plasma membrane disruption; ventral disc fragmentation	[112]
Beta-lapachone	Nuclei Vacuoles Plasm membrane	chromatin condensation; vacuolization; membrane blebbing	[113]
[(10H-phenothiazin-10-yl)methyl]-N-hydroxybenzamide	Vacuoles Cytokinesis	Myelinic figures; multinucleated parasites	[116]
Furazolidone	Cytoplasm contents Vacuoles	depleted cytoplasm; lamellar structures	[107]
Bisphosphonates	Vacuoles	Myelinic figures	[118]
24,25-(R,S)-epiminolanosterol	Vacuoles Peripheral Vesicles	Myelinic figures; peripheral vesicle increased	[117]
Nocodazole	Flagella Ventral disc Median Body	flagellar length increased; ventral disc disrupted; median body reduced	[119,120]
Colchicine	Ventral disc Mitosis	ventral disc disrupted; cytokinesis blocked	[120]
Taxol	Flagella Median Body	flagellar length decreased; median body increased	[120]
Oryzalin	Flagella	flagella bending	[121]
Polo-like kinases specific inhibitor	Flagella	Flagella length increased	[122]
Cytochalasin D	Flagella Cytokinesis Peripheral vesicles (endocytosis)	flagella internalization multiflagellated unshaped cells; ceramide absorption reduced	[125,131]
Pyrantel pamoate	Flagella Peripheral vesicles	flagella beating decreased peripheral vesicle increased	[124]
Small gatekeeper kinases and aurora kinases Kinase inhibitors	Cytokinesis	multinucleated parasites	[126,127]
3-arylideneindolin-2-one-type sirtuin inhibitor	Nuclei Vacuoles	DNA fragmentation Myelin figures	[9]

## 11. Ribosomes

The ribosome is a universal structure synthesizing protein in all cells. In the case of prokaryotic cells, it turns out to be one important drug target. Indeed, 40% of the presently used antibiotics have the ribosome as the main target. Although the ribosomes of eukaryotic cells look identical, cryo-electron microscopy of isolated ribosomes from various cell types has noted some particularities that may represent new drug targets. Previous studies have shown that *Giardia* ribosomes differ from prokaryote and eukaryote cells. It presents the shortest rRNA in eukaryotic cells and is even shorter than in the prokaryote counterparts [134]. Cryo-EM analysis revealed that the *Giardia* ribosome is an extremely reduced version missing all the eukaryotic-specific rRNA expansion segments. This observation opens the possibility of designing specific *Giardia* ribosome inhibitors [134].

Another potential approach in the chemotherapy against Giardiasis is to consider signaling pathways involved in the occurrence and development of inflammation, a process in which prostaglandins play a key role. One good example is the role played by cyclooxygenase-2 (COX-2). Indeed, a group of 2H indazole derivatives was designed and synthesized, and one of them (No.18) shows a much higher activity against *Giardia* than metronidazole. This compound is also an inhibitor of COX-2 [135].

## 12. Conclusions

To improve the quality of the chemotherapy against parasitic protozoa, it is important to invest in the development of highly specific compounds that interfere with key steps of essential metabolic pathways or to disassemble functional macromolecular complexes, most of the time associated with cell structures and organelles. Therefore, it is especially important to incentivize further metabolic studies based on molecular analysis and proofs of concept based on the blockage of synthesis of certain proteins using gene silencing, CRISP-Cas9, etc. The data presented in this review point to potential targets that still require further study.

## Figures and Tables

**Figure 4 microorganisms-10-02176-f004:**
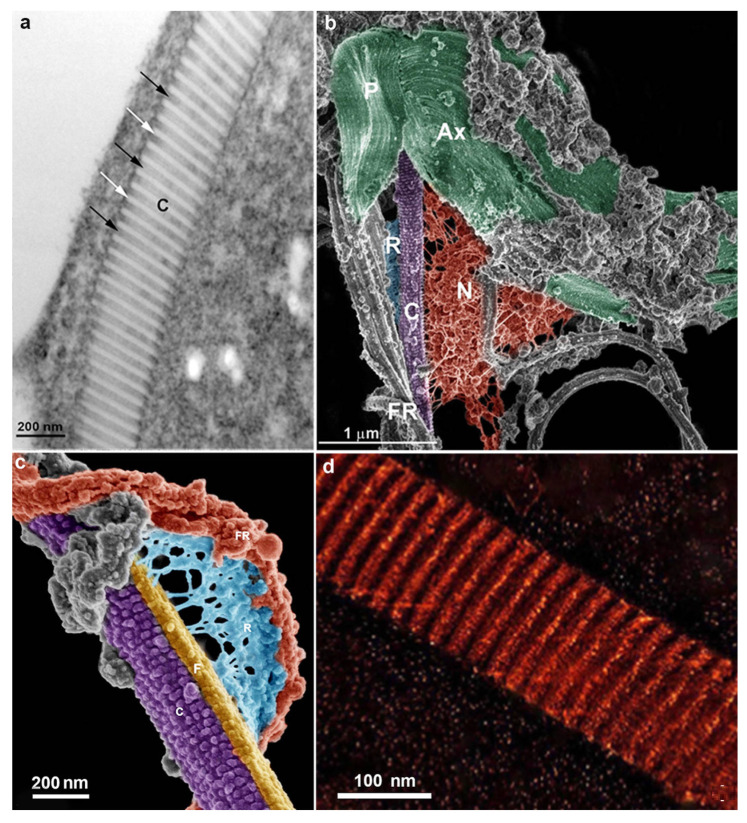
The costa observed under different methodologies: (**a**) by transmission electron microscopy. Arrows point to electron dense bands with regular periodicity; (**b**) high-resolution scanning electron microscopy (XHR-SEM) of the striated fibers of Trichomonas after plasma membrane remotion with detergent. Ax, axostyle, Costa (C), P, Pelta; N, nucleus; (**c**) XHR-SEM of the filamentous network (R), connecting the accessory filament (F) to the recurrent flagellum (FR); and (**d**) three-dimensional reconstruction of the isolated costa by electron tomography of the isolated costa show electron-lucent and electron-dense bands. After Refs. [29,30].

**Figure 5 microorganisms-10-02176-f005:**
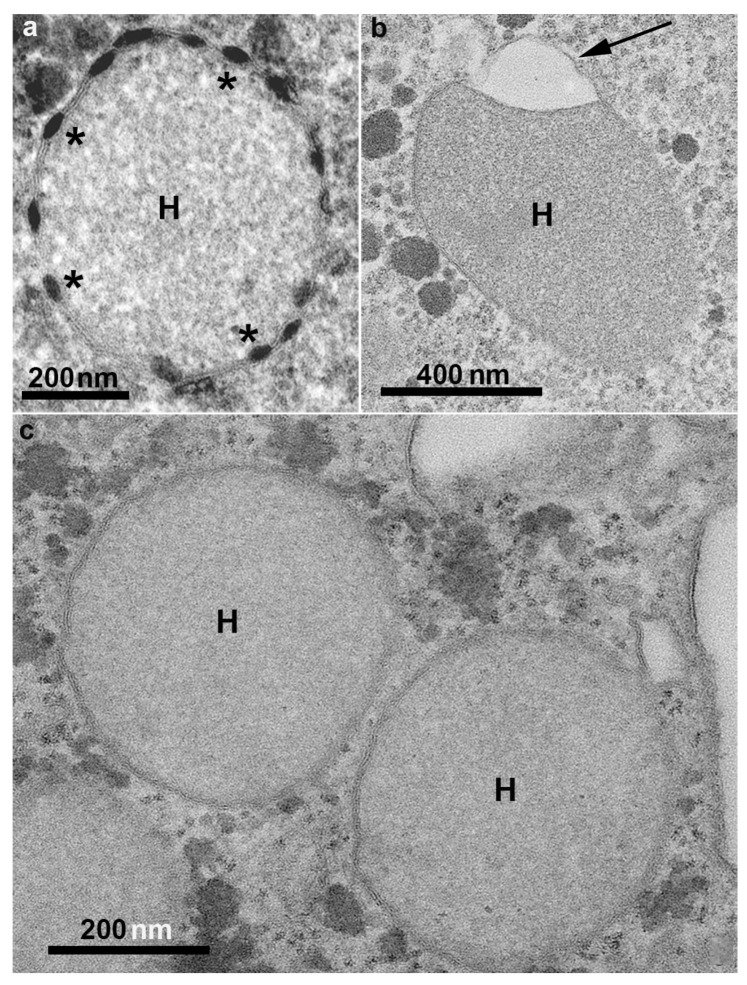
Hydrogenosomes (H) of *T. vaginalis* observed by transmission electron microscopy. The organelle can present several peripheral vesicles (asterisks) with electron density (**a**); or only one flat vesicle (arrow in (**b**)); notice the double membrane surrounding the organelles (**c**). (Benchimol, unpublished).

**Figure 6 microorganisms-10-02176-f006:**
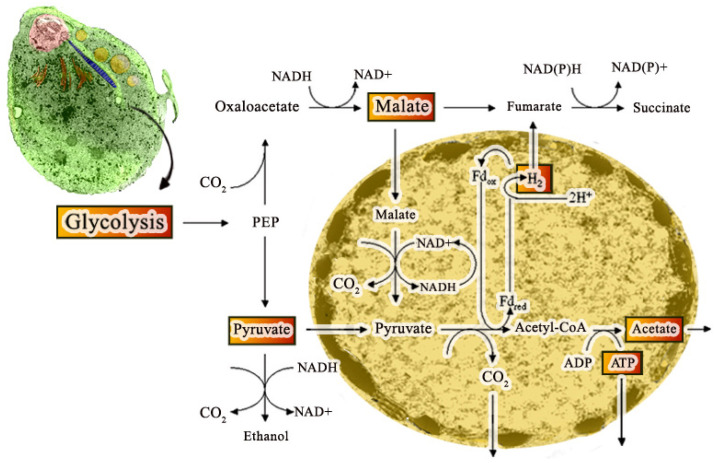
Scheme of the metabolic cycle that occurs in the hydrogenosome. After glycolysis in the cytosol, pyruvate enters the hydrogenosome. Malate is also utilized after its decarboxylation to pyruvate. Within the hydrogenosome, pyruvate is oxidatively decarboxylated to acetyl coenzyme A and CO_2_, catalyzed by an iron-sulfur protein, PFOR. Reduced ferredoxin is reoxidized by hydrogenase in a reaction that reduces protons to molecular hydrogen. As a result, ATP and acetate are produced. (After Ref. [25]).

**Figure 7 microorganisms-10-02176-f007:**
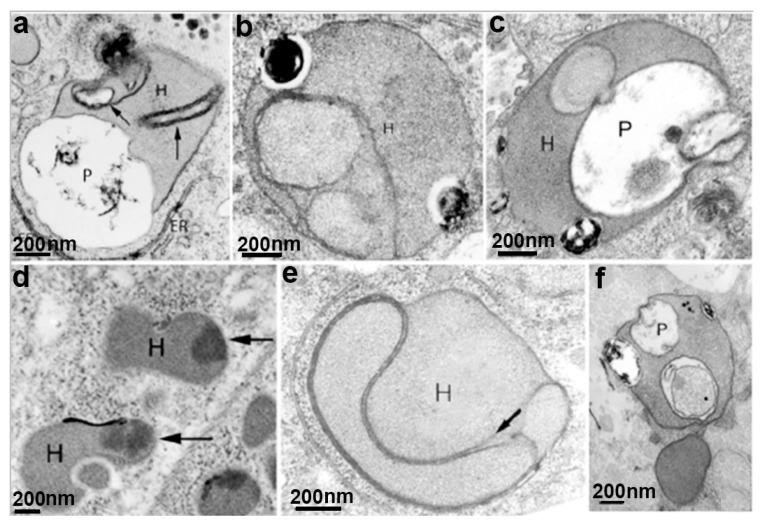
Hydrogenosomes observed after drug treatments: (**a**) with hydroxyurea, (**b**,**c**) metronidazole; (**d**) zinc; (**e**) cytochalasin; and (**f**) fibronectin. The hydrogenosomes are giant, with internal compartments and abnormal peripheral vesicles (arrows). H, hydrogenosomes; P, pelta; ER, endoplasmic reticulum. (After Ref. [25]).

**Figure 8 microorganisms-10-02176-f008:**
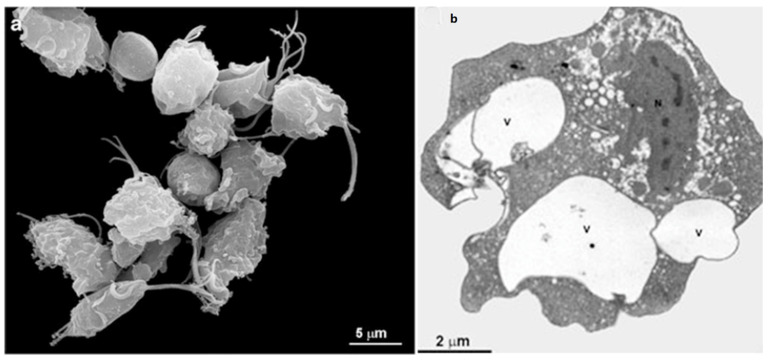
Miltefosine treatment. Parasites were treated with 14.5 μM Mitelfosine for 24 h and observed with SEM (**a**); and TEM (**b**). By SEM, cells are rounded (57%) and wrinkled (52%) with membrane blebbing (42%); (**b**) an intense vacuolization (V) and nuclear condensation were observed (After ref. [57]).

**Figure 9 microorganisms-10-02176-f009:**
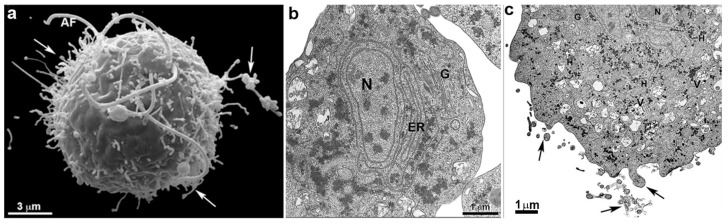
Parasites treated with clotrimazole (CTZ) and zinc compounds were observed by SEM (**a**) and TEM (**b**,**c**). Membrane projections (filopodia) are visible (arrows) (**a**,**c**), which are enhanced with [Zn(CTZ)2(Ac)2] treatment. By TEM, note an atypical ER with a high number of filopodia (arrows) and an irregular distribution of Golgi lamellas after treatment. AF, anterior flagela. N, nucleus, G, Golgi complex, ER, endoplasmic reticulum. (After Ref. [61]).

**Figure 10 microorganisms-10-02176-f010:**
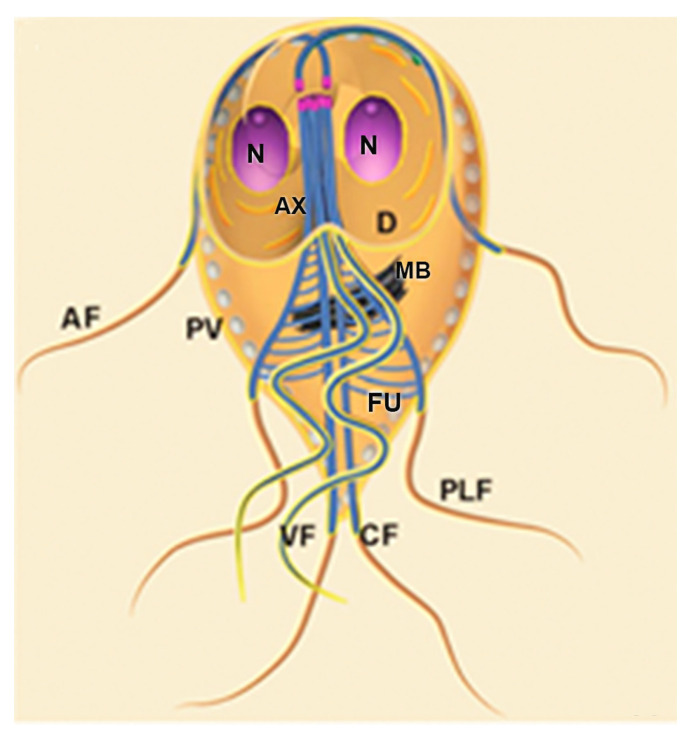
Scheme of *G. intestinalis*. The trophozoite displays two nuclei (N), peripheral vesicles (PV), ventral disc (D), median body (MB), funis (Fu), and four flagella pairs (AF—anterior flagella; VF—ventral flagella; PLF—postero-lateral flagella; CF—caudal flagella). (After Ref. [78]).

**Figure 11 microorganisms-10-02176-f011:**
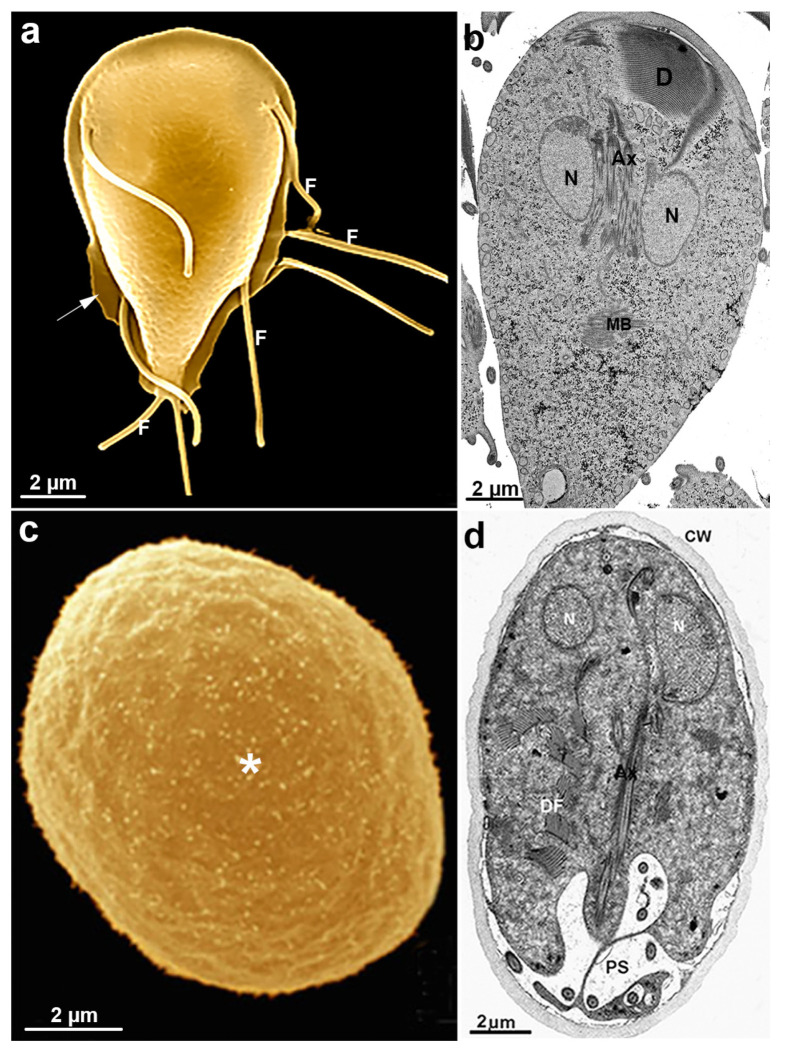
*G. intestinalis* trophozoite and cyst: (**a**) scanning electron microscopy of the parasite. The four flagella (F) pairs, ventrolateral flange (arrow), and dorsal surface are observed; (**b**) transmission electron microscopy of a trophozoite showing internal structures as the nuclei (N), median body (MB), ventral disc (D), and axonemes (Ax); (**c**) scanning electron microscopy image shows an oval appearance of the cyst and a fibrillar material (*) deposited over the cell; and (**d**) transmission electron microscopy of the cyst. The nuclei (N), disc fragments (DF), and axonemes (Ax) are observed, as well as the cyst wall (CW) and peritrophic space (PS). (**a**,**b**) (Gadelha and Benchimol, unpublished); (**c**,**d**) (After Ref. [78]).

**Figure 12 microorganisms-10-02176-f012:**
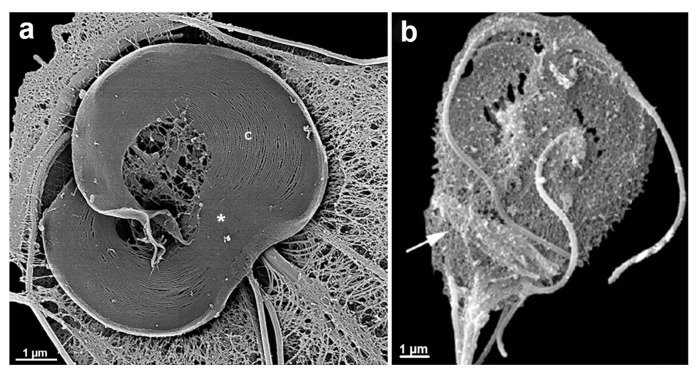
*G. intestinalis* cytoskeleton: (**a**) the ventral disc (D) is seen by Helium Ion Microscopy. Disc microtubules are arranged in a spiral; and (**b**) scanning electron microscopy of *G. intestinalis* in dorsal view showing the median body (arrow). C, costa; asterisks, basal bodies. (**a**) (After Ref. [79]); (**b**) (After Ref. [75]).

**Figure 14 microorganisms-10-02176-f014:**
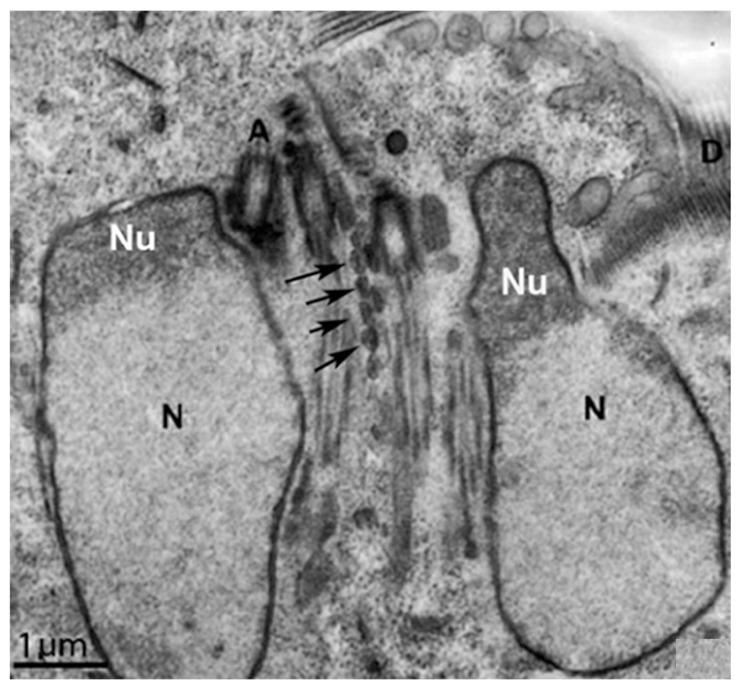
*G. intestinalis* nuclei have an oval aspect and are localized in the anterior region. A thin section (transmission electron microscopy) of a trophozoite cut parallel to the surface of the ventral disc (D). Note the proximity of the axonemes (A) of the anterior, postero-lateral, and ventral flagella to the nuclei (N). The nucleolus (Nu) is also observed. Central mitosomes (arrows) are seen between the axonemes. (After Ref. [73]).

**Figure 15 microorganisms-10-02176-f015:**
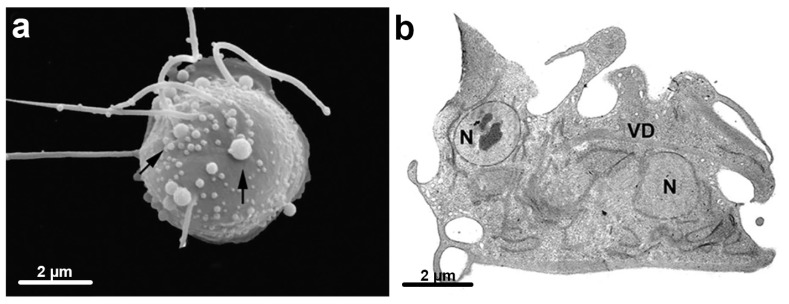
*G. intestinalis* after beta-lapachone treatment: (**a**) scanning electron microscopy of treated trophozoite shows membrane blebbing (arrows); and (**b**) transmission electron microscopy of the trophozoite shows condensed chromatin in the nucleus. (After Ref. [113]).

**Figure 16 microorganisms-10-02176-f016:**
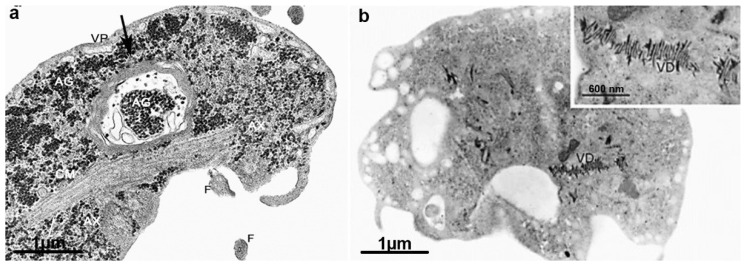
Transmission electron microscopy of *G. intestinalis* trophozoites after drug exposure: (**a**) vacuole with myelinic figures and glycogen granules (arrow) is observed after treatment with 4-[(10H-phenothiazin-10-yl)methyl]-N-hydroxybenzamide; and (**b**) cells treated with nocodazole display ventral disc (VD) fragmentation (inset). (**a**) (After Ref. [116]); (**b**) (After Ref. [119]).

**Figure 17 microorganisms-10-02176-f017:**
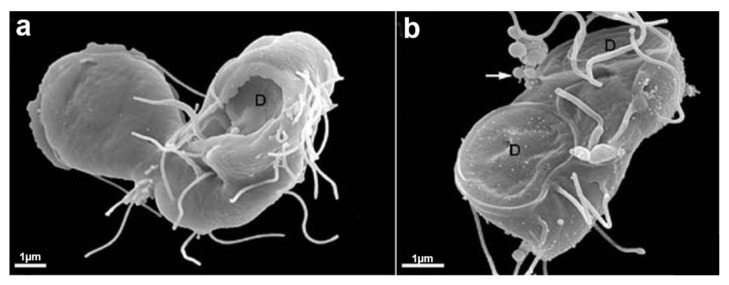
Scanning electron microscopy of *G. intestinalis* after 10 μM cytochalasin treatment. (**a**,**b**) Multiflagellar cells are observed. D, ventral disc. (After Ref. [125]).

**Figure 18 microorganisms-10-02176-f018:**
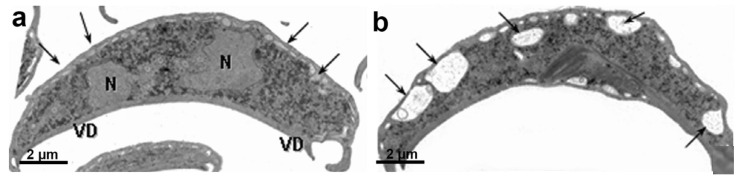
Transmission electron microscopy of *G. intestinalis* trophozoites: (**a**) the control parasite shows two nuclei (N), peripheral vesicles (arrows), and a ventral disc (VD); and (**b**) parasite after treatment with 24,25-(R,S)-epiminolanosterol. Notice the peripheral vesicles enlargement (arrows). (After Ref. [117]).

## Data Availability

Not applicable.

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
