# Peer review of "Unusual Cell Structures and Organelles in Giardia intestinalis and Trichomonas vaginalis Are Potential Drug Targets"

_microorganisms, 2022, doi:10.3390/microorganisms10112176_

Round 1

Reviewer 1 Report

The review paper presented seems to be a very good work, which can help researchers to search for new active compounds against the written targets and therefore contribute to the development of antiparasitic drugs, not against the two parasites that they describe, but against others , while following their reasoning. However, in this review and although I know that it is difficult to collect all the information, some references to new active synthesis compounds against Trypanosoma cruzi and Trichomonas, which have been patented in Spain and whose activity and selectivity indices are superior to reference drugs. However, I reiterate that the work is very well structured, especially the part and the pharmacological targets

Author Response

Referee 1: The review paper presented seems to be a very good work, which can help researchers to search for new active compounds against the written targets and therefore contribute to the development of antiparasitic drugs, not against the two parasites that they describe, but against others, while following their reasoning.

However, in this review and although I know that it is difficult to collect all the information, some references to new active synthesis compounds against Trypanosoma cruzi and Trichomonas, which have been patented in Spain and whose activity and selectivity indices are superior to reference drugs. However, I reiterate that the work is very well structured, especially the part and the pharmacological targets.

Answer: Thanks for the suggestion. The suggested reference was included in the manuscript and table 1.

Ibáñez-Escribano, A., Fonseca-Berzal, C., Martínez-Montiel, M., Álvarez-Márquez, M., Gómez-Núñez, M., LacuevaArnedo, M., Espinosa-Buitrago, T., Martín-Pérez, T., Escario, J.A.,  Merino-Montiel, P., Montiel-Smith, S., Gómez-Barrio, A., López, O., Fernández-Bolaños, J.G. Thio- and selenosemicarbazones as antiprotozoal agents against Trypanosoma cruzi and Trichomonas vaginalis, J Enzyme Inhibit Med Chem 2022. 37:1, 781-791. https://doi.org/10.1080/14756366.2022.2041629.

Reviewer 2 Report

This is a very interesting paper on the unusual structure of two important parasites and the potential of these structures as drug targets. The paper does need some serious editorial help regarding English sentence structure (on page 2, the sentence “the targets may include structures and organelles…” is not clear and very difficult to understand), misspellings, and formatting errors.

All figures have high quality and present rich details.

The introduction is short and presents only classification information about Giardia intestinalis. Why this information about Trichomonas vaginalis is lacking? Why this information is relevant to this manuscript?

The section “chemotherapy against parasitic protozoan” has two paragraphs with no references, mainly in the paragraph where the authors present data from different approaches to developing new chemotherapies to treat both infections.

In the section “trichomonas vaginalis features”, the authors describe the signals and symptoms of trichomoniasis in women and men, however, this section is incomplete and much information is missing. For example, the author mentioned infertility in women because of the infection; however, it can occur in men. This section should be reviewed.

The manuscript does not bring new information or a new interpretation of previous studies. There is no originality in this manuscript. The presented information is relevant, but they were published in previous reviews. The authors mentioned the mechanism of action of metronidazole and the possible resistance mechanism and cited the effects of other drugs on Trichomonas vaginalis and Giardia intestinalis, but these data were previously published and no new issues are discussed. I expected a discussion about old drugs' action on new targets, but I did not see this discussion. Instead, I saw old data.

Some new compounds showed anti-Trichomonas and anti-Giardia activity, however, there was no discussion about toxicity to human cells. Many new compounds present potent in vitro activity, but the toxicity is very high.

Is necessary a discussion put all data together and point out how the old approved drugs can be used to treat both infection and possible action mechanisms. Is necessary a discussion about the action on the exclusive structures. The way the data are presented is only speculative.

Author Response

Referee 2: This is a very interesting paper on the unusual structure of two important parasites and the potential of these structures as drug targets. The paper does need some serious editorial help regarding English sentence structure (on page 2, the sentence "the targets may include structures and organelles…" is not clear and very difficult to understand), misspellings, and formatting errors.

Answer: Many thanks. On page 2, the sentence was changed to: "Some structures and organelles, such as hydrogenosomes in trichomonads and the ventral disc in Giardia, are fundamental for the parasite's survival. Thus, drugs that can interfere with the functions played by these structures and organelles may constitute new alternatives for treating the diseases caused by this pathogenic protist."

Referee: All figures have high quality and present rich details. The introduction is short and presents only classification information about Giardia intestinalis. Why this information about Trichomonas vaginalis is lacking? Why this information is relevant to this manuscript?

Answer: Thanks. It was included.

Referee: The section "chemotherapy against parasitic protozoan" has two paragraphs with no references, mainly in the paragraph where the authors present data from different approaches to developing new chemotherapies to treat both infections.

Answer: Both paragraphs deal with general statements related to the area of chemotherapy that does not require any specific reference.

In the section "trichomonas vaginalis features, "the authors describe the signals and symptoms of trichomoniasis in women and men; however, this section is incomplete and much information is missing. For example, the author mentioned infertility in women because of the infection; however, it can occur in men. This section should be reviewed.

Answer: This is not the main purpose of this review to deal with the clinical aspects of giardiasis and trichomoniasis. We only briefly reference this point and refer to other more specific articles.

Referee: The manuscript does not bring new information or a new interpretation of previous studies. There is no originality in this manuscript. The presented information is relevant, but they were published in previous reviews.

Answer: This is a review paper based on previously published papers. None of the previous reviews correlated drug targets with the structural organization of the two protists discussed here.

Referee: The authors mentioned the mechanism of action of metronidazole and the possible resistance mechanism and cited the effects of other drugs on Trichomonas vaginalis and Giardia intestinalis, but these data were previously published, and no new issues were discussed.

Answer: As commented above, this is a review manuscript where no original data are incorporated.

Referee: I expected a discussion about old drugs' action on new targets, but I did not see this discussion. Instead, I saw old data.

Answer: The main purpose of this review was not to center on the mechanisms of action of available drugs but to indicate structures and organelles that may constitute potential drug targets.

Referee: Some new compounds showed anti-Trichomonas and anti-Giardia activity; however, there was no discussion about toxicity to human cells. Many new compounds present potent in vitro activity, but the toxicity is very high.

 Answer: This review did not intend to center on the toxicity of the mentioned drugs. It is well known that in the early phase of drug development against parasites, attention is given to identifying compounds that can kill them. In the second phase, we deal with the important question of host toxicity, where chemical changes can be made to decrease toxicity while maintaining the anti-parasite activity.

Referee: Is necessary a discussion put all data together and point out how the old approved drugs can be used to treat both infection and possible action mechanisms. It is necessary a discussion about the action on the exclusive structures. The way the data are presented is only speculative.

Answer: Again, this review manuscript mainly focused on the presentation of structures and organelles of two pathogenic protisits that may constitute new drug targets. This is not an original manuscript.

Reviewer 3 Report

The manuscript focuses on the very precise description of cell structures, organelles and biochemical processes of Giardia intestinalis and Trichomonas vaginalis as a potential target for a drug.

In my opinion the review is well written and a very big advantages are the great images.

Furthermore, it is an excellent resource that provides all the basic information in one place for readers interested in new drug design against giardiasis and trichomonosis.

However, I have one issue for the authors: could the authors increase the section 10.1 “Metronidazole, Albendazole and Nitazoxanide”? which are still the most common used drugs. Apart from this one suggestion, I have nothing to add.

Author Response

Referee 3

The manuscript focuses on the very precise description of cell structures, organelles and biochemical processes of Giardia intestinalis and Trichomonas vaginalis as a potential target for a drug. In my opinion the review is well written and a very big advantages are the great images.

Furthermore, it is an excellent resource that provides all the basic information in one place for readers interested in new drug design against giardiasis and trichomonosis.

However, I have one issue for the authors: could the authors increase section 10.1 "Metronidazole, Albendazole and Nitazoxanide"? which are still the most common used drugs. Apart from this one suggestion, I have nothing to add.

Answer: We added more comments on the compounds mentioned.

We hope we have addressed all relevant questions raised by the reviewers.

Round 2

Reviewer 2 Report

There is no suggestion. I accepted all comments and the justification of the authors.